# Determining responsiveness and meaningful changes for the Musculoskeletal Health Questionnaire (MSK-HQ) for use across musculoskeletal care pathways

Andrew James Price [1], Reuben Ogollah,[2] Sujin Kang,[1] Elaine Hay,[2] Karen L Barker,[1] Elena Benedetto,[1] Stephanie Smith,[1] James Smith,[1] James B Galloway,[3] Benjamin Ellis,[4] Jonathan Rees,[1] Sion Glyn-Jones,[1] David Beard,[1] Ray Fitzpatrick,[5] Jonathan C Hill[2]

For numbered affiliations see end of article.

**Correspondence to**
Dr Andrew James Price;
andrew.price@ndorms.ox.ac.uk

## ABSTRACT

**Objectives** We have previously developed and validated the Arthritis Research UK Musculoskeletal Health Questionnaire (MSK-HQ) for use across musculoskeletal care pathways, showing encouraging psychometric test results. The objective of this study was to determine the responsiveness of MSK-HQ following MSK treatments and to determine the minimally important change (MIC).

**Setting** We collected data in four cohorts from community physiotherapy and secondary-care orthopaedic hip, knee and shoulder clinics.

**Participants** 592 individuals were recruited; 210 patients treated with physiotherapy for a range of MSK conditions in primary care; 150 patients undergoing hip replacement, 150 patients undergoing knee replacement and 82 undergoing shoulder surgery in secondary care.

**Outcome measures** Preoperative data were collected including the MSK-HQ, European Quality of Life-5D (EQ-5D) and the OHS, OKS or OSS in each joint-specific group. The same scores, together with anchor questions, were collected postintervention at 3 months for the physiotherapy group and 6 months for all others. Following COnsensus-based Standards for the selection of health status Measurement INstruments (COSMIN) guidelines, responsiveness was assessed using correlation between scores and the MIC was calculated for the entire cohort using receiver operating characteristic curve analysis.

**Results** The MSK-HQ demonstrated strong correlation (R=0.73) with EQ-5D across the entire cohort and with each of the joint-specific Oxford scores (hip R=0.87, knee R=0.92 and shoulder R=0.77). Moderate correlation was seen between MSK-HQ and EQ-5D across each individual group (R value range 0.60–0.68), apart from the hip group where correlation was strong (R=0.77). The effect size with MSK-HQ was 0.93, in the entire cohort, double that measured with EQ-5D (0.43). In all subgroups, MSK-HQ measured a greater treatment effect compared with EQ-5D. The MIC is 5.5 (95% CI 2.7 to 8.3).

## Strengths and limitations of this study

► The Musculoskeletal Health Questionnaire (MSK-HQ) has been shown to be responsive across a range of musculoskeletal (MSK) conditions.
► The MSK-HQ performed well in comparison with other quality-of-life-based questionnaires, including the European Quality of Life-5D score.
► The minimally important change value for the MSK-HQ in the general MSK population has been calculated.
► A limitation of the study is that the MSK-HQ has not been tested across all MSK conditions.

**Conclusion** Our study demonstrates that the MSK-HQ questionnaire is responsive to change across a range of musculoskeletal conditions, supporting its use as a generic MSK measurement instrument.

## INTRODUCTION

Musculoskeletal (MSK) health problems affect one in four of the adult population and managing this diverse spectrum of conditions is expensive, for instance in the National Health Service (NHS) accounting for approximately £5 billion of spending annually.[1] Patients with MSK pain usually enter the clinical pathway via their primary care doctor presenting with a wide range of different diagnoses and representing up to 14% of all consultations.[2] Given the scale of the MSK healthcare burden, new ways of providing care are developing across all healthcare systems. Within the UK, the pathway for providing MSK services has developed significantly over the last 15 years.[3] Primary care treatment is effective

in managing many MSK problems but a significant proportion of patients will be referred through an intermediate care pathway where more diagnostic tests may be undertaken and physiotherapy care provided.[4] A proportion of these patients have symptoms resistant to initial therapy and are referred on to hospital-based secondary care for assessment by orthopaedic, rheumatology or sports medicine doctors.[5] These pathways have become formally established within the UK with the development of integrated MSK clinical assessment and treatment services.[6] The development of care pathways in MSK services has occurred in parallel with the drive towards validated outcome-based assessment of the care provided and value-based commissioning.[7] To facilitate this, there has been a requirement to create a relevant generic MSK patient-reported outcome measure (PROM) that could be used to assess all MSK patients as they initially enter and then progress through the pathway, regardless of diagnosis.[8] This requirement reflects the increasing need across all healthcare models to embed the measurement of clinical change over time. To achieve this, a responsive PROM is required. The European Quality of Life Questionnaire (EQ-5D-5L) patient measure has been previously widely used for this role in MSK care, but some concerns have been raised regarding its lack of responsiveness.[9 10] This may reflect the measurement properties of EQ-5D-5L as a generic healthcare quality of life measure rather than a condition or disease-specific measure. More specifically, the EQ-5D-5L individual items have no musculoskeletal context for patients, which is an important characteristic for embedding them in a clinical pathway.

To address this need, the Arthritis Research UK Musculoskeletal Health Questionnaire (MSK-HQ) was developed.[8] This is a generic MSK PROM, designed to be used across the full spectrum of MSK conditions. The MSK-HQ contains 14 items capturing symptoms and functional problems that patients with MSK conditions have identified as important. These include pain severity, physical function, work interference, social interference, sleep, fatigue, emotional health, physical activity, independence, understanding, confidence to self-manage and overall impact. We have previously shown that the MSK-HQ has encouraging psychometric test results with high completion rates (94%), excellent test-retest reliability (intraclass correlation coefficient=0.84) and a high level of internal consistency (Cronbach's α=0.88).[8] In addition, the score demonstrates very good convergent validity when measured against EQ-5D with a Spearman rank correlation coefficient of 0.81, respectively.[8]

The specific aim of this study was to determine the responsiveness of the MSK-HQ in measuring clinical change as a single generic MSK measure following treatment across different MSK pathways. In addition, the study aimed to determine the minimally important change (MIC) for MSK-HQ.

## METHODS
### Cohorts
We have developed a series of cohorts of patients to validate the MSK-HQ as previously described.[8] We used follow-up outcome data from the cohorts to determine the MSK-HQ's responsiveness and MIC, with data collected longitudinally at two time points.

### Community physiotherapy cohort
Patients were recruited from community MSK physiotherapy clinics in five UK West Midlands towns (Middlewich, Congleton, Wombourne, Cheadle and Wolverhampton), where NHS treatment is offered after referral by the General Practitioner (GP). Patients were invited to take part and recruited directly in the clinic having received a study information pack. The inclusion criteria were age >18 years and referral from a GP for physiotherapy treatment. No further inclusion/exclusion criteria were used, creating a study group with a heterogeneous range of unspecified MSK problems. Patients consented to fill out questionnaires pretreatment and at 3 months after first consultation.

### Secondary care orthopaedic cohorts (knee, hip, shoulder)
From the Nuffield Orthopaedic Centre in Oxford, we recruited separate knee, hip and shoulder cohorts from patients who had been listed for surgery. Patients were approached in their preassessment clinic appointment prior to surgery having received a study information pack. Inclusion criteria were age >18 years and patient listed for (1) knee replacement surgery, (2) hip replacement surgery and (3) shoulder surgery including open and arthroscopic procedures (excluding instability surgery). Patients consented to fill out a preoperative questionnaire and again at 6-month postsurgery. The shoulder cohort had been expanded to 82 compared with the 60 previously reported.[8]

### Outcome measures
All participants were asked to complete the MSK-HQ and EQ-5D-5L (the primary reference measure) as baseline and final follow-up as the second time point. The MSK-HQ is a 14-item questionnaire assessing pain severity, physical function, work interference, social interference, sleep, fatigue, emotional health, physical activity, independence, understanding, confidence to self-manage and overall impact. When reporting the MSK-HQ, all 14 items are summed together (responses coded from 'not at all'=4 to 'extremely'=0, except for items 12 and 13, which have the response options in the reverse order) providing a range from 0 to 56, with higher scores indicating better MSK health status.[8] The EQ-5D-5L questionnaire is a generic quality of life measure, which consists of five domains (mobility, self-care, usual activities, pain/discomfort and anxiety/depression) with five categorical levels for each domain. In addition, the instrument contains a general health VAS (0–100, 0=best health) item. EQ-5D-5L can be interpreted in the form of an index score or in the form

of 243 unique health states. In this study, the EQ-5D-5L utility score was calculated using the UK Crosswalk value set, with scores ranging between –0.59 and 1 (full health).[11] In addition, the Orthopaedic Cohorts filled out a joint specific; Oxford Knee Score (OKS), Oxford Hip Score (OHS) or Oxford Shoulder Score (OSS) as joint-specific reference measures. The OKS, OHS and OSS questionnaires follow are PROMs with 12-item addressing pain and functional. In each Oxford questionnaire, every item has five answer categories in a Likert scale, with items now scored from 0 to 4. The summary score range in all scores is 0 (worst) to 48 (best).[12–14] The EQ-5D-5L, OKS, OHS and OSS are all widely used PROMs with and we have previously published a detailed analysis of their established measurement properties.[15]

This was a longitudinal study and in each group the same scores were recorded at final follow-up (3 months for physiotherapy and 6 months for orthopaedic cohorts) corresponding to the normal clinic practice in each clinical setting. The questionnaires are all patient reported and were filled out by the patients with no assistance from the research or clinical teams.

### Evaluation of responsiveness and calculation of MIC

For each of the prospectively collected cohorts, we compared the baseline and follow-up scores (3 months for the physiotherapy cohort and 6 months for orthopaedic cohorts). There is no gold standard measurement tool in this context and so we could not apply area under the curve methodology. Following COnsensus-based Standards for the selection of health status Measurement INstruments (COSMIN) guidelines, we therefore used the construct approach and we set a priori hypotheses of expected positive associations at the moderate or above level between MSK-HQ and the reference measurement tools (EQ-5D-5L, OKS, OHS, OSS) (table 1). To test these, we examined the magnitude and direction of change, calculating the Spearman correlation coefficients (non-parametric data) for comparison of MSK-HQ and other measures, within the different groups. The strength of the positive or negative correlation coefficients was interpreted according to Munro little, if any (0.00–0.25), low (0.26–0.49), moderate (0.50–0.69), high (0.70–0.89) and very high correlation (0.90–1.00).[16] We expected the correlations to be positive and to be moderate or above.

We then performed subgroup analysis calculating and comparing the standardised effect size (SES—the mean change score divided by the SD of the measure at baseline) and the standardised response mean (SRM—the mean change score divided by the SD of the change scores) for the outcome measure used within each cohorts collected.

For the entire cohort, the clinically relevant change (improvement or deterioration) was determined by calculating the MIC using receiver operating characteristic (ROC) curve analysis. For the external anchor, we used the following transition item from the Outcome

| Table 1 | The eight a priori hypotheses that were examined | |
|---|---|---|
| Number | Hypotheses | Result |
| 1 | The correlation between the MSK-HQ and EQ-5D within the entire cohort is moderate (r≥0.50) or above | Accepted |
| 2 | The correlation between the MSK-HQ and EQ-5D within the physiotherapy cohort is moderate (r≥0.50) or above | Accepted |
| 3 | The correlation between the MSK-HQ and EQ-5D within the knee cohort is moderate (r≥0.50) or above | Accepted |
| 4 | The correlation between the MSK-HQ and EQ-5D within the hip cohort is moderate (r≥0.50) or above | Accepted |
| 5 | The correlation between the MSK-HQ and EQ-5D within the shoulder cohort is moderate (r≥0.50) or above | Accepted |
| 6 | The correlation between the MSK-HQ and OKS within the knee cohort is moderate (r≥0.50) or above | Accepted |
| 7 | The correlation between the MSK-HQ and OHS within the hip cohort is moderate (r≥0.50) or above | Accepted |
| 8 | The correlation between the MSK-HQ and OSS within the knee cohort is moderate (r≥0.50) or above | Accepted |

EQ-5D-5L, The European Quality of Life Questionnaire; MSK-HQ, Musculoskeletal Healthcare Questionnaire; OHS, Oxford Hip Score; OKS, Oxford Knee Score; OSS, Oxford Shoulder Score; r, Spearman correlation coefficient.

and Experience Questionnaire to identify change in clinical state compared with baseline (Question: 'How would you now rate the problem you recently came to clinic or hospital for?' Response: 'Much better', 'a little better', 'the same', 'a little worse' or 'much worse').[17] For calculation of the MIC, the responses were dichotomised using two response groups: (1) improvement=a little better and (2) no improvement=same.

### COSMIN guidelines

We followed COSMIN guidelines for evaluation of responsiveness and calculation of MIC.[18] The percentage of missing items is reported. Complete case analyses were performed throughout the analyses for the MSK scores, with no imputation for missing values. The power calculation for sample size of the cohorts has been described in first publication regarding these study groups based on evaluating test-retest properties.[8] Using COSMIN standards, the sample size for follow-up data was judged as very good for the total group, the physiotherapy, hip and knee cohorts (seven times number of items or ≥100 subjects).[19] The sample size for the shoulder cohort was inadequate.

**Table 2** Demographic characteristics of patients recruited across the four cohorts

| Patient characteristics | All participants | Physiotherapy | Hip | Knee | Shoulder |
|---|---|---|---|---|---|
| Number | 592 | 210 | 150 | 150 | 82 |
| Age (years), mean (SD) | 56.0 (16.9) | 53.3 (15.5) | 55.6 (17.2) | 65.7 (13.8) | 53.0 (16.7) |
| Female, n (%) | 326 (55.1) | 112 (53.3) | 88 (58.7) | 89 (59.3) | 37 (45.1) |
| Male, n (%) | 266 (43.5) | 98 (46.7) | 62 (41.3) | 61 (40.7) | 45 (54.9) |

## Statistical analysis

All analyses were conducted in STATA/IC V.14 (StataCorp LP, 2015), SPSS V.22 (IBM Corp, 2013) and Statistical software-R V.3.2.2 (The R Foundation for Statistics, 2015).

## Patient and public involvement

The MSK-HQ questionnaire in a PROM and its development has involved patients in each of the following stages: development of the research question, development of the outcome measures itself and participating in the research.

## RESULTS
## Demographic data

There were 592 patients in total who consented to participate in the four cohort studies (210 physiotherapy patients, 150 hip, 150 knee, 82 shoulder). Baseline population characteristics for the overall sample and by each cohort are summarised in table 2, showing a mean age of 56 years (SD 16.9) and with a split of 56.5% female and

43.5% male. As expected, the symptom episode duration was much greater in the secondary care cohorts compared with the community physiotherapy cohort.

## Descriptive analysis of baseline and follow-up scores

Table 3 demonstrates that across the entire cohort there was an improvement in mean baseline score to follow-up score for both MSK-HQ and EQ-5D. A very similar pattern was seen for the physiotherapy cohort. In the joint-specific cohorts, the same trend occurred across all three scores used in each setting. The response rate for the entire cohort was 70%.

## Responsiveness

There was a strong correlation between change scores for MSK-HQ and non-disease-specific EQ-5D across the entire cohort with an R value of 0.73 (table 4). There was moderate correlation between the same measures in the physiotherapy (0.67), knee (0.67) and shoulder cohorts (0.57), with strong correlation in the hip cohort (0.72). When comparing change scores between MSK-HQ and

**Table 3** Base line and follow-up outcomes scores for each cohort

| Cohort | Surveys for each cohort [Lower–upper range} | Baseline Mean | SD | Follow-up* Mean | SD | Follow-up response rate (%) |
|---|---|---|---|---|---|---|
| Total | MSK-HQ (0, 56) | 28.6 | 9.6 | 41.3 | 11.4 | 70 |
| (n=592) | EQ-5D-5L (−0.59, 1) | 0.5 | 0.3 | 0.7 | 0.2 | |
| Physiotherapy | MSK-HQ | 30.5 | 9.6 | 40.0 | 11.3 | 63 |
| (n=210) | EQ-5D-5L (−0.59, 1) | 0.6 | 0.3 | 0.7 | 0.2 | |
| Hip | MSK-HQ | 24.9 | 8.3 | 43.0 | 12.0 | 71 |
| (n=150) | EQ-5D-5L (−0.59, 1) | 0.4 | 0.2 | 0.7 | 0.3 | |
| | OHS (0, 48) | 20.4 | 8.6 | 37.4 | 10.0 | |
| Knee | MSK-HQ | 27.5 | 9.0 | 40.8 | 11.5 | 82 |
| (n=150) | EQ-5D-5L (−0.59, 1) | 0.5 | 0.3 | 0.7 | 0.2 | |
| | OKS (0, 48) | 20.9 | 8.8 | 34.6 | 10.1 | |
| Shoulder | MSK-HQ | 32.1 | 10.4 | 42.9 | 9.4 | 44 |
| (n=82) | EQ-5D-5L (−0.59, 1) | 0.6 | 0.3 | 0.7 | 0.2 | |
| | OSS (0, 48) | 29.6 | 10.3 | 37.5 | 7.4 | |

Missing data for baseline MSK-HQ: all participants n=50 (8.2%), physiotherapy cohort n=5 (2.4%), hip cohort n=4 (2.7%), knee cohort n=22 (14.7%),shoulder cohort n=2 (2.4%).
*Follow-up time point: physiotherapy cohort=3 months, hip/knee/shoulder cohorts=6 months.
EQ-5D-5L, The European Quality of Life Questionnaire (range −0.59 and 1); MSK-HQ, Musculoskeletal Healthcare Questionnaire; OHS, Oxford Hip Score (range 0–48);OKS, Oxford Knee Score (range 0–48); OSS, Oxford Shoulder Score (range 0–48).

**Table 4** Correlations in change score between MSK-HQ and other reference outcome scores (EQ-5D, OHS, OKS and OSS) in the entire group and the cohorts (physiotherapy, hip, knee and shoulder)

| Instrument | Comparator | Cohort | Spearman correlation coefficient (r) |
|---|---|---|---|
| MSK-HQ | EQ-5D | Total | 0.73 |
| MSK-HQ | EQ-5D | Physiotherapy | 0.67 |
| | | Hip | 0.77 |
| | | Knee | 0.68 |
| | | Shoulder | 0.60 |
| MSK-HQ | OHS | Hip | 0.87 |
| | OKS | Knee | 0.92 |
| | OSS | Shoulder | 0.77 |

EQ-5D-5L, The European Quality of Life Questionnaire ; MSK-HQ, Musculoskeletal Healthcare Questionnaire; OHS, Oxford Hip Score; OKS, Oxford Knee Score; OSS, Oxford Shoulder Score.

**Table 5** Effect size and standard response mean for MSK-HQ and other established outcome score in the four cohorts (physiotherapy, hip, knee and shoulder)

| Cohort (number responding) | | Effect size (95% CIs)* | SRM (95% CIs)† |
|---|---|---|---|
| Physiotherapy (n=147) | MSK-HQ | 0.93 (0.66 to 1.20) | 0.99 (0.71 to 1.26) |
| | EQ-5D-5L | 0.43 (0.18 to 0.68) | 0.46 (0.21 to 0.71) |
| Knee (n=107) | MSK-HQ | 1.53 (1.18 to 1.86) | 1.05 (0.74 to 1.35) |
| | OKS | 1.52 (1.19 to 1.85) | 1.26 (0.95 to 1.57) |
| | EQ-5D-5L | 0.94 (0.65 to 1.23) | 0.87 (0.58 to 1.15) |
| Hip (n=123) | MSK-HQ | 2.14 (1.73 to 2.53) | 1.56 (1.21 to 1.90) |
| | OHS | 1.93 (1.53 to 2.33) | 1.59 (1.22 to 1.95) |
| | EQ-5D-5L | 1.21 (0.88 to 1.52) | 1.03 (0.72 to 1.34) |
| Shoulder (n=36) | MSK-HQ | 1.05 (0.52 to 1.57) | 1.08 (0.55 to 1.60) |
| | OSS | 0.73 (0.22 to 1.23) | 0.99 (0.45 to 1.51) |
| | EQ-5D-5L | 0.73 (0.24 to 1.20) | 0.76 (0.29 to 1.24) |

*Using baseline (BL) (SD).
†Using paired differences (SD).
EQ-5D-5L, The European Quality of Life Questionnaire; MSK-HQ, Musculoskeletal Healthcare Questionnaire; OHS, Oxford Hip Score; OKS, Oxford Knee Score; OSS, Oxford Shoulder Score.

Oxford scores in the joint-specific cohorts, the correlations for OKS (0.93) and OHS (0.82) were particularly strong, with moderate correlation seen with the OSS (0.61). The correlations meant in all cases the a priori hypotheses were accepted (table 1).

Across all cohorts, the MSK-HQ measured a relatively large mean effect size for treatment ranging between 2.14 in the hip cohort and 0.93 in the physiotherapy cohort. Within the physiotherapy cohort, greater responsiveness was shown with MSK-HQ compared with EQ-5D, with over twice the effect size measured at 3 months. In the knee cohort, MSK-HQ measured a similar effect size to the OKS (1.53 and 1.52, respectively) at 6 months, both of which were greater than that calculated using EQ-5D (0.94). A similar pattern was seen within the hip cohort. Within the shoulder cohort, the pattern was different with MSK-HQ measuring a greater effect size than OSS and EQ-5D, which delivered similar results across the cohort (both 0.73). The findings were supported by the calculated standardised response mean values (table 5).

### Minimally important change
The MIC for MSK-HQ was 5.5 (95% CI 2.7 to 8.3) based on the optimal cut-off value with specificity 0.66 and sensitivity 0.65, with area under the ROC curve 0.66 (see figure 1).

### DISCUSSION
The results of this study demonstrate that the MSK-HQ PROM is responsive to change across patients with a range of MSK conditions. When considering change scores across the entire cohort, the MSK-HQ performed well compared with EQ-5D with a strong correlation between the scores. Within the subgroups, the MSK-HQ correlated very well with change measured using the OHS, OKS and OSS. This was supported by measured effect sizes,

particularly for OKS and OHS. For the community physiotherapy group, the correlation with EQ-5D was moderate but there was a much larger measured effect size using the MSH-HQ. The trend for the new score to measure a larger effect size when compared with EQ-5D was seen across all the cohorts.

The requirement to integrate PROMs into clinical management and outcome-based commissioning in primary and intermediate care has expanded.[7] The MSK-HQ PROM was designed to be applicable across MSK conditions and to be meaningful to patients particularly in the primary and intermediate care setting.[8] Therefore, it is important to reflect on the results of the MSK-HQ across the whole mixed cohort and the community physiotherapy cohort, which does represent a truly mixed population of different MSK conditions. Previous work has shown that the MSK-HQ has excellent test-retest reliability and strong convergent validity with reference standards.[8] Results from the entire cohort and the physiotherapy cohort suggest that the MSK-HQ is responsive to change in the general MSK population and is applicable to the primary and intermediate care setting. Considering its use in this setting, we have calculated a single MIC value for the MSK-HQ for application across all MSK conditions. The MIC value represents the smallest change in MSK-HQ score that represents a meaningful change in symptoms over time for an individual patient within a care pathway, a score that has been calculated for many other outcome measures.[20] Physiotherapy has a very important role to play in the initial non-operative management of nearly all MSK conditions, and for

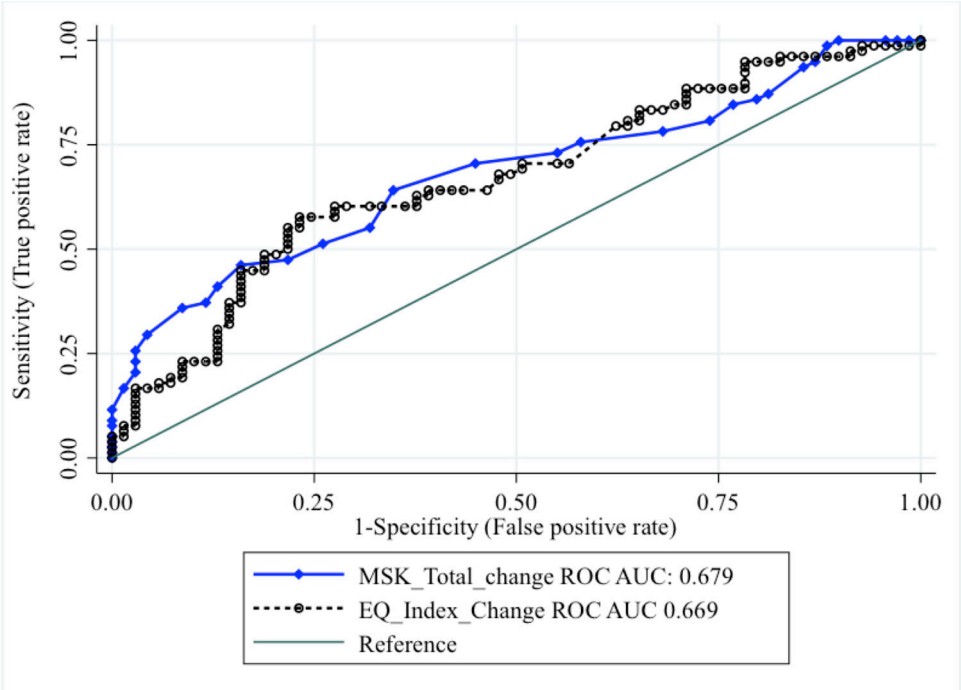

**Figure 1** A graph showing the receiver operating characteristic (ROC) analysis using MSK-HQ and EQ-5D index change scores across the entire cohort. AUC, area under the ROC curve; MSK, Musculoskeletal Healthcare Questionnaire.

many years the outcome measure of choice in this clinical space has been EQ-5D.[21–25] A key vision of the MSK-HQ was to fill the current gap for a single broad health status measure instead of relying on generic health tools such as the EQ-5D-5L, which have been shown to be less sensitive to change in MSK populations.[9 10] One key requirement for the MSK-HQ yet to be tested is whether it is more sensitive to change than the EQ-5D-5L. Encouragingly, the MSK-HQ performed favourably in comparison to EQ-5D in terms of responsiveness to change across the cohort.

Despite the MSH-HQ not being joint or condition specific, the new score performed well in responsiveness when compared with the OHS, OKS and OSS. In particular, the score appears to measure change in response to hip and knee arthroplasty in a similar manner to the OKS and OHS with very high measured correlation coefficients and similar effect sizes. The follow-up sample size for the shoulder cohort was inadequate, and further work is required to determine responsiveness in this group of patients.

Our study does have a number of limitations that must be addressed. We have not collected any information regarding detailed nature of pathology or treatment within each cohort, such as stage of disease, time to treatment or detail of reconstructive surgery. Our aim was not to explore these issues but only to measure change in the MSK-HQ before and after treatment to calculate responsiveness across the whole MSK cohort. Our study does have missing data from the follow-up cohort. However, overall the effective response rate for the entire cohort was 70%, which is considered acceptable.[26] In calculating

responsiveness, the sample size for the entire group and all individual cohorts, other than shoulder, was very good as judged against COSMIN standards. Although the study was performed solely in the NHS and the study population did not cover all MSK diagnoses, we believe the study group is representative and the results generalisable to the broad group of MSK patients who exist in all healthcare systems.[8]

In the our previous publication regarding the production of MSK-HQ, we laid out a set of prerequisites that should be met for its creation.[8] We believe that our previous findings together with the results of this study show that these conditions have now been met. The score has been coproduced with patients and clinicians and reflects aspects of health that were meaningful to both. It can provide an MSK-specific quantification of a person's MSK health. The score can be used across the MSK pathway with different MSK conditions and treatment targets. It does demonstrate robust psychometric properties and it is responsive to change enabling longitudinal measurement and the monitoring of changes over time. In this study, we have demonstrated that it is applicable for use by different MSK health professionals and feasible for use in routine clinical practice across the MSK patient pathway.

The expansion of MSK intermediate care within the NHS has highlighted the need for a generic PROM to assist in developing outcome-based commissioning of care. The measurement property profile of the MSK-HQ PROM appears to support its use in this area, although its role is still evolving. Increasingly, there is a common clinical pathway for most MSK patients within the NHS.

Patients initially seek advice and treatment from their GP and if their condition does not settle are referred to an MSK intermediate care referral hub, which acts as a gateway to secondary care. The pathway is not joint or condition specific, so a generic MSK PROM to measure clinical change and response to treatment is required for the process. The MSK-HQ appears to demonstrate the appropriate measurement properties and responsiveness to fulfil this role.

## CONCLUSION

In summary, the findings from this study show that the MSK-HQ appears to be responsive to change across a range of MSK conditions, performing favourably when compared with EQ-5D.

**Author affiliations**
[1]Nuffield Department of Orthopaedics, Rheumatology and Musculoskeletal Sciences, University of Oxford, Oxford, UK
[2]Institute of Primary Care and Health Sciences, Keele University, Stoke on Trent, UK
[3]Departmant of Rheumatology, Kings College London, London, UK
[4]Arthritis Research UK, London, UK
[5]Nuffield Department of Population Health, University of Oxford, Oxford, UK

**Acknowledgements** The authors thank the clinical teams and patients that participated in the study. The authors acknowledge support from the Oxford NIHR Biomedical Research Centre.

**Contributors** AJP, RF, BE, DB, EH, JBG and JCH contributed to the conception and design of the work. JCH, EH, SS, SK, EB, JS, KLB, DB, JR, SG-J and AJP were involved in running the cohorts and collecting data. RO, AJP, SK and JCH were involved in the analyses and AJP, JCH and RF in the interpretation of the data. AJP, RO, EH, KLB, SS, JS, JBG, JR, SG-J, DB, EB, RF, JCH and BE were involved in the drafting of the manuscript and its revision for important intellectual content and gave final approval for the manuscript.

**Funding** This project was funded by Arthritis Research UK (ref 20518). AJP is part funded through the Oxford NIHR Biomedical Research Centre.

**Competing interests** None declared.

**Patient consent for publication** Not required.

**Ethics approval** Ethical approval for this study was obtained through the IRAS system (IRAS ref 168971) via Wales REC 4 committee (REC ref 15/WA/0040).

**Provenance and peer review** Not commissioned; externally peer reviewed.

**Data availability statement** Data are available upon reasonable request.

**ORCID iD**
Andrew James Price http://orcid.org/0000-0002-4258-5866

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
