## [Reviewer comments · BMJ Open]

ARTICLE DETAILS

TITLE (PROVISIONAL)	Determining Responsiveness and Meaningful Changes for the Musculoskeletal Health Questionnaire (MSK-HQ) for use across musculoskeletal care pathways.
AUTHORS	Price, Andrew; Ogollah, Reuben; Kang, Sujin; Hay, Elaine; Barker, Karen L.; Benedetto, Elena; Smith, Stephanie; Smith, James; Galloway, James B; Ellis, Benjamin; Rees, Jonathan; Glyn-Jones, Sion; Beard, David; Fitzpatrick, Ray; Hill, Jonathan

VERSION 1 – REVIEW

REVIEWER	Florian Baumann Department of Trauma Surgery Regensburg University Medical Center, Germany
REVIEW RETURNED	13-Aug-2018

GENERAL COMMENTS	The manuscript entitled "Determining Responsiveness and Meaningful Changes for the Musculoskeletal Health Questionnaire (MSK-HQ) for use across musculoskeletal care pathways." is a prospective, controlled, non-randomized clinical trial presenting mid-term longitudinal data on the MSK-HQ. The study is based on the population from the initial validation study in 2016. The study reports data to complete validation data on psychometric properties of the MSK-HQ and is therefore of potential interest for publication. However, there are some limitations that need to be addressed. Introduction: The introduction is well written. However, the argumentation is very specific to the NHS environment and NHS pathways, which suggests a limitation concerning generalizability. The authors should explain why the EQ-5D fails to respond to clinically relevant changes (global health scale vs. disease-specific health scale). M&M Did you have any a-priori hypotheses? If so, please state. There is no determination of the stage of the disease except the duration of symptoms (e.g. radiographic changes (Kellgren-Lawrence score), ROM, leg axis in knee OA). Did you ask for any history of trauma to evaluate any rapid-progressive degenerative conditions due to post-traumatic OA? How much time was between the indication for arthroplasty and surgery (not the global duration of symptoms)? In other words, how long were the patients listed before surgery? You report on missing data. Please state if there was any cluster in missing data, any evidence for a misunderstanding of any question? Typing error p9l15 COSMIN checklist. Discussion
---

	The discussion is relatively short and lacking information on the advantages/disadvantages of a disease-specific scale compared to the previously used GH-QoL scale. What are the special requirements for a questionnaire in musculoskeletal disorders? Please discuss why you chose different periods of follow-up for the study groups. Typing error p13,l50. Please add a paragraph on limitations of the study including the above-mentioned issues (generalizability, determination of the stage of the disease) Please revise the whole manuscript for comma rules.
--	--

REVIEWER	Rawan AlHeresh MGH Institute of Health Professions USA
REVIEW RETURNED	21-Sep-2018

GENERAL COMMENTS	The objective of this study was to evaluate the responsiveness of the Musculoskeletal Health Questionnaire (MSK-HQ) following treatment as well as to determine minimally important change associated with this patient-reported outcome. This study needs improvement for the following reasons:  1. In the introduction section, the authors mention the EQ-5D. There is no prior introduction for this abbreviation or what it measures? Please add further information about the “construct” you are trying to tackle in your argument. The argument for needed yet another PRO is not there at all, how is the MSK-HQ better and why? 2. Towards the end of page 6 in the introduction, please list numeric values for the “encouraging” psychometric test results you are referring to. 3. The methods section is very brief and lacks a lot of detail. Was ethics approval granted? How were people recruited? Did the participants consent? Who completed the assessments, did the physician ask the patients to complete the forms? All the outcome measures lack detail about the constructs they measure, scoring method, and psychometric qualities. Why would you choose shoulder and hip specific measures despite having participants in the sample that do not have issue with these joint specifically? 4. Methods> evaluating responsiveness section. I am assuming you mean the COSMIN guidelines? Some introduction on these standards are needed (also please include the full name before the abbreviation). 5. Numbers and percentages in table 1. contain some errors that need altering. 6. Table 2. could be presented better (PT versus Ortho outcomes) to emphasize that the follow up is at different points. In the footnote please add score ranges and what they represent. 7. Methods> responsiveness section. Per the authors mention of using the COSMIN guidelines; I was surprised to see confidence intervals listed in table 3. I also do not see any mention of the direction and magnitude of expected correlation versus what they really have. Please revise according to the COSMIN guidelines. 8. I could not see a figure one for the MIC.
---

VERSION 1 – AUTHOR RESPONSE

Reviewer: 1

Reviewer Name: Florian Baumann

Institution and Country: Department of Trauma Surgery, Regensburg University Medical Center, Germany

Please state any competing interests or state 'None declared': I have no competing interests to declare.

Please leave your comments for the authors below:

The manuscript entitled "Determining Responsiveness and Meaningful Changes for the Musculoskeletal Health Questionnaire (MSK-HQ) for use across musculoskeletal care pathways." is a prospective, controlled, non-randomized clinical trial presenting mid-term longitudinal data on the MSK-HQ. The study is based on the population from the initial validation study in 2016. The study reports data to complete validation data on psychometric properties of the MSK-HQ and is therefore of potential interest for publication. However, there are some limitations that need to be addressed.

Introduction:

The introduction is well written. However, the argumentation is very specific to the NHS environment and NHS pathways, which suggests a limitation concerning generalizability.

The authors should explain why the EQ-5D fails to respond to clinically relevant changes (global health scale vs. disease-specific health scale).

The text has been changed to acknowledge that this is a global healthcare issue. We have tried to use the UK as an example of the issue hence making the paper more generalizable.

We have changed the text offering some explanation as to the responsiveness of the EQ-5D measure – as suggested.

M&M : Did you have any a-priori hypotheses? If so, please state.

This study did have number of a-priori hypotheses that examined and these have now been listed in the Table 2.

There is no determination of the stage of the disease except the duration of symptoms (e.g. radiographic changes (Kellgren-Lawrence score), ROM, leg axis in knee OA). Did you ask for any history of trauma to evaluate any rapid-progressive degenerative conditions due to post-traumatic OA? How much time was between the indication for arthroplasty and surgery (not the global duration of symptoms)? In other words, how long were the patients listed before surgery?

The study recruited 4 different cohorts of patients. (1) Those referred to a physiotherapy led clinic (multiple MSK conditions) all treated with non-operative care (representing a general MSK diagnosis clinic), (2) Those undergoing Orthopaedic intervention in secondary care, recruited as separate hip, knee and shoulder cohorts. Within this no history of the specific condition was recorded or presented. The study was not designed to record or report differences in history (e.g post-traumatic OA) or the time between listing and surgery. The text has been adjusted to reflect this in the methods section and the issue addressed in the limitations section of the discussion.

You report on missing data. Please state if there was any cluster in missing data, any evidence for a misunderstanding of any question?

We have reported in detail regarding missing data in the first publication regarding this questionnaire in these same cohorts (see reference below). This is now more explicitly referenced in the text. The analysis shows no significant clustering.

Hill JC, Kang S, Benedetto E, et al. Development and initial cohort validation of the Arthritis Research UK Musculoskeletal Health Questionnaire (MSK-HQ) for use across musculoskeletal care pathways. *BMJ Open* 2016; 6:e012331.doi:10.1136/bmjopen-2016-012331

Typing error p9l15 COSMIN checklist.

Error corrected

Discussion: The discussion is relatively short and lacking information on the advantages/disadvantages of a disease-specific scale compared to the previously used GH-QoL scale. What are the special requirements for a questionnaire in musculoskeletal disorders?

The discussion has been revised to reflect the reviewer's comments. We have put forward the specific requirements for a questionnaire in musculoskeletal disorders and illustrated how the MSQ-HQ addresses them.

Please discuss why you chose different periods of follow-up for the study groups?

The follow up periods are different for each study group but are consistent within each group. This reflects normal clinical practice within each cohort. This has now been emphasised in the text and limitations paragraph.

Please add a paragraph on limitations of the study including the above-mentioned issues (generalizability, determination of the stage of the disease)

A paragraph on limitations has been expanded to include comments about generalizability and stage of disease.

Please revise the whole manuscript for comma rules.

We have edited the text.

Reviewer: 2

Reviewer Name: Rawan AlHeresh

Institution and Country: MGH Institute of Health Professions, USA

Please state any competing interests or state 'None declared': None declared

Please leave your comments for the authors below : The objective of this study was to evaluate the responsiveness of the Musculoskeletal Health Questionnaire (MSK-HQ) following treatment as well as to determine minimally important change associated with this patient-reported outcome. This study needs improvement for the following reasons:

1. In the introduction section, the authors mention the EQ-5D. There is no prior introduction for this abbreviation or what it measures? Please add further information about the “construct” you are trying to tackle in your argument. The argument for needed yet another PRO is not there at all, how is the MSK-HQ better and why?

Thank you for your comments. We have previously published the first validation paper regarding the MSK-HQ in the BMJ-Open and this paper details the argument as to why another PRO was required. We have edited the text of this paper to refer to the requirement for the MSK-HQ referring to the first paper. We have added more detail describing the EQ-5D.

2. Towards the end of page 6 in the introduction, please list numeric values for the “encouraging” psychometric test results you are referring to.

We have listed the values for completion rate (%), test-retest reliability (Intraclass Correlation Coefficient), internal consistency (Cronbach’s) and convergent validity (Pearson’s and Spearman’s rank correlation coefficients).

3. The methods section is very brief and lacks a lot of detail.

Was ethics approval granted?

Ethical approval was granted. This was entered into the manuscript in the after the end of the main manuscript as is requested in the BMJ formatting advice. In addition it has been added to the methods section

How were people recruited?

More detail has been added to the text regarding patient recruitment.

Did the participants consent?

All patients gave their consent to be involved in the study and this has been acknowledged in the text.

Who completed the assessments; did the physician ask the patients to complete the forms?

The assessments were all made using patient reported outcome measures, where the questionnaires are given to the patient by a researcher for baseline measurement and then posted to the patient for repeat measurements. The patient was not assisted in filling out the questionnaires. This has been clarified in the text.

All the outcome measures lack detail about the constructs they measure, scoring method, and psychometric qualities.

We have added detail regarding all the outcome measures used in the study.

Why would you choose shoulder and hip specific measures despite having participants in the sample that do not have issue with these joint specifically?

There are 4 distinct groups of patients in this study. (1) A physiotherapy group with different MSK pathologies involving different joints where MSK-HQ and EQ-5D measures were used (2) A knee cohort where MSK-HQ, EQ-5D and OKS were used, (3) A hip cohort where MSK-HQ, EQ-5D and

OHS were used and (4) A shoulder cohort where MSK-HQ, EQ-5D and OSS were used. Therefore we have only used joint specific scores in cohorts where the pathology is joint specific. We apologise if this was not clear in the text and we have edited the manuscript to emphasise this.

4. Methods> evaluating responsiveness section. I am assuming you mean the COSMIN guidelines? Some introduction on these standards are needed (also please include the full name before the abbreviation).

We have added text around the COSMIN standards. We have stated the full name before the abbreviation.

5. Numbers and percentages in table 1. contain some errors that need altering.

Table 1 has been amended so that numbers and percentages are now correct.

6. Table 2. Could be presented better (PT versus Ortho outcomes) to emphasize that the follow up is at different points. In the footnote please add score ranges and what they represent.

Table 2 has been amended to reflect the point above. The differences in follow up are highlighted in the text.

7. Methods> responsiveness section. Per the authors mention of using the COSMIN guidelines; I was surprised to see confidence intervals listed in table 3. I also do not see any mention of the direction and magnitude of expected correlation versus what they really have. Please revise according to the COSMIN guidelines.

The text has been revised accordingly.

8. I could not see a figure one for the MIC.

Figure 1 is included in the revised manuscript.

VERSION 2 – REVIEW

REVIEWER	Florian Baumann Department of Trauma Surgery, Regensburg University Medical Center
REVIEW RETURNED	22-Mar-2019
GENERAL COMMENTS	To my opinion the changes are an improvement of the manuscript. I recommend acceptance.